# Causal Structure Learning for Sparse Matrix Fill-in Reduction

**Ziwei Li** [1 2]  **Shuzi Niu** [1 3]  **Tao Yuan** [1 3]  **Huiyuan Li** [1 3]

## Abstract

The performance of sparse direct solvers is fundamentally governed by fill-in, i.e. new nonzero entries arising from the LU factorization of a sparse matrix, as they dictate memory footprint and subsequent computation time. For decades, a variety of graph-theoretic algorithms have aimed to minimize fill-in, a problem known to be both NP-hard and critically important. While recent deep learning methods, optimizing surrogate fill-in objectives, show empirical promise and can outperform classical algorithms on certain matrices, they offer limited interpretability into the underlying mechanism of fill-in generation. To address this, we propose a novel reordering approach, Causal Triplet Structure Learning (CTS), which is grounded in the Fill-Path Theorem and reduces arbitrary-length fill-paths to length-two candidate triplets, identifies the causal structures that trigger fill-in, and intervenes to block their formation. Empirically, we design a multigrid-style GAT with KAN activations to learn vertex embeddings and introduce a causal triplet loss that discourages such structures during training. Experiments on the SuiteSparse Matrix Collection demonstrate that our method reduces fill-in by $6\times$, leading to $12\times$ speedup in factorization time compared to state-of-the-art methods on Chemical Process Simulation and Computational Fluid Dynamics matrices.

## 1. Introduction

Sparse direct solvers remain a cornerstone for large-scale sparse linear systems arising in scientific and engineering

---

[1]Institute of Software, Chinese Academy of Sciences, Beijing, China [2]University of Chinese Academy of Sciences, Beijing, China [3]Key Laboratory of System Software, Institute of Software, Chinese Academy of Sciences, Beijing, China. Correspondence to: Shuzi Niu <shuzi@iscas.ac.cn>, Huiyuan Li <huiyuan@iscas.ac.cn>, Tao Yuan <yuantao@iscas.ac.cn>.

*Proceedings of the $43^{rd}$ International Conference on Machine Learning*, Seoul, South Korea. PMLR 306, 2026. Copyright 2026 by the author(s).

simulations, such as finite element analysis, computational fluid dynamics, circuit simulation, and optimization (Wilkinson & Reinsch, 1971; George & Liu, 1981; Duff et al., 1986; Saad, 2003). They are particularly important when high accuracy is required, the system is nearly singular or ill-conditioned, or many right-hand sides must be handled efficiently through reuse of a single factorization (Davis et al., 2016). In such solvers, runtime and memory consumption are governed not only by floating-point operations but also by fill-in, i.e., additional nonzero entries that are created in the factors during Gaussian elimination or LU factorization (Liu, 1990). To control the fill-in explosion that multiplies nonzeros hundreds-fold, matrix reordering ultimately determines whether factorization takes hours or weeks at a cost of less than 1% of time. For matrices of size around 10,000, the LU factorization via SuperLU (splu) takes minutes. With a suitable matrix reordering, this computational time can be cut by an order of magnitude.

Fill-in minimizing matrix reordering is NP-hard yet crucial. Classical graph-theoretic algorithms include locally greedy minimum-degree variants (e.g., AMD) (Amestoy et al., 1996; Davis et al., 2004), separator-based nested dissection (ND) (George, 1973) to confine fill-in propagation, and spectral ordering using the Fiedler vector (Barnard et al., 1993) to cluster connected nodes. While interpretable, their generalization is limited. Deep learning approaches like AlphaElimination (Dasgupta & Kumar, 2023) using CNN-enhanced MCTS for global elimination decisions, UDNO (Li et al., 2026) concentrating nonzeros near the diagonal, and PFM (Li et al., 2025) minimizing nonzeros via differentiable relaxation offer better generalization. All these methods attempt to optimize fill-in-related objective functions in the training process, but how their prediction results are related to fill-ins remains unclear.

In terms of cause-and-effect relationships, fill-ins seem to be the effect of LU factorization over a sparse matrix. The elimination of each row or column is implemented through elementary row operations in the context of Gaussian elimination/LU decomposition or vertex elimination in the graph-theoretic interpretation. According to the Fill-Path Theorem (Rose et al., 1976), fill-in is created during elimination: even if two vertices are not adjacent in the original graph, a fill edge between them can be induced if there exists a path connecting them whose intermediate

vertices are all eliminated earlier than the two endpoints. Theoretically, we only need to enumerate and block all the fill-paths of length 2 to minimize fill-ins. Those reduced fill-paths are candidate causal structures and finally determined with elimination ordering. Here we explore how to learn to identify those causal structures for fill-in reduction.

We introduce Causal Triplet Structure Learning (CTS), a learning framework designed to discover causal structures that facilitate fill-in optimization. For prediction, we employ a multigrid-style message-passing backbone, enhanced by a novel Graph Attention Network (Veličković et al., 2018) with Fourier-KAN activations as the vertex embedding module. The proposed architecture is particularly well-suited for objectives with mathematical structure. From the vertex embeddings, we derive an elimination criterion. Candidate triplets that satisfy the "intermediate-elimination-first" inequality give rise to causal triplet structures, which correspond directly to fill-ins in the factorized matrix. For optimization, we design a causal triplet loss function that discourages the formation of such causal structures, thereby enforcing compliance with the inequality condition and reducing fill-in. Extensive experiments on benchmark sparse Matrix Collection SuiteSparse (Davis & Hu, 2011) with matrix size ranging from 10,000 up to the largest size permitted by device memory demonstrate that the proposed method CTS consistently reduces fill-in and translates these structural improvements into measurable speedups in LU factorization. The predicted causal structures are evaluated with de facto fill-ins through SciPy's splu (Python Wrapper of SuperLU) (Li, 2005). Our method reduces fill-ins by $6\times$, leading to a factorization acceleration of $12\times$ compared to state-of-the-art methods.

The key contributions are summarized as follows: (1) *Fill-Paths as Causal Structures.* We theoretically reduce the arbitrary-length fill-generation paths based on Fill-Path theorem to minimal causal triplets of length 2. These condensed structures explicitly encode elimination-order information and serve as the direct causes of fill-in. (2) *Causal Learning Formulation.* We reframe the fill-reduction task as a causal-structure identification problem and introduce a novel causal triplet loss that directly optimizes the elimination order to avoid fill-inducing causal configurations. (3) *Interpretable Neural Ordering Model.* A multilevel message-passing architecture is built on a multigrid backbone with each level modeled by a novel Graph Attention Network with Fourier-KAN activations. This is designed to capture mathematically structured patterns for vertex-elimination prediction. (4) *Empirical Superiority.* The approach achieves state-of-the-art performance on sparse matrix benchmarks, significantly reducing fill-in while also accelerating the end-to-end LU factorization time.

## 2. Related Work

Sparse matrix reordering is traditionally studied through a graph interpretation. For a sparse matrix, rows and columns can be viewed as vertices, and nonzero entries define edges in the adjacency graph (Harary, 1969; Rose, 1972). Under this correspondence, many effective heuristics have been developed over the past decades. Early bandwidth- and profile-oriented methods such as Cuthill–McKee (Cuthill & McKee, 1969) and its reverse variant RCM reduce the envelope by performing level-structure traversals (George, 1971). To target fill-in during factorization more directly, minimum-degree (MD) and modern variants such as AMD iteratively eliminate low-degree vertices in the evolving elimination graph, offering strong practical performance with low overhead (Amestoy et al., 1996). Separator-based strategies, most notably nested dissection (ND), recursively bisect the graph with small separators to expose block structure and enable scalable fill-reducing orderings (George, 1973). In practice, high-quality implementations such as METIS compute ND-style orderings efficiently via multilevel coarsening/refinement (Karypis & Kumar, 1998). Beyond these, spectral techniques based on sorting the components of a Laplacian eigenvector (often the Fiedler vector) provide an alternative global criterion and have been used for envelope reduction and related ordering tasks (Barnard et al., 1993). Despite their maturity and widespread adoption, no single heuristic is uniformly optimal across matrix families, as performance depends strongly on sparsity structure and problem characteristics.

For sparse reordering, a key difficulty is that the fill-in count produced by exact sparse factorization is a discrete, permutation-dependent quantity that is revealed only after (symbolic/numeric) elimination, making it hard to formalize into a tractable and differentiable training objective. Consequently, most learning-based approaches optimize objectives that are easier to learn, while aiming to correlate with reduced fill-in in downstream factorization. Deep reinforcement learning is a natural choice to formulate matrix reordering as a sequential decision problem. DRL_ND (Gatti et al., 2022) learns to construct separators/partitions in a multilevel framework, using objectives such as normalized cut or vertex-separator quality as training signals. AlphaElim (Dasgupta & Kumar, 2023) instead treats the matrix as an image and models the numeric elimination process, using Monte Carlo tree search guided by a neural policy to choose elimination actions.

More recently, end-to-end learning frameworks have attempted to tighten the connection between learned scores and factorization outcomes. UDNO (Li et al., 2026) learns vertex scores and encourages locality in the resulting permutation (placing adjacent vertices close in the ordering), which empirically shrinks the set of potential fill locations

by promoting near-diagonal concentration. PFM (Li et al., 2025) goes one step further by bringing a factorization procedure into the optimization loop. It minimizes an $\ell_1$-based proxy defined on the triangular factors computed within training, rather than on the exact factors, and uses reparameterization together with proximal-style optimization to connect vertex scores, permutations, and factor-related objectives. While these objectives can be effective in practice, they are still surrogate or approximate with respect to the discrete fill-in generated by exact elimination, and they typically do not encode the structural rule by which fill edges are created. This observation motivates our theory-guided approach, which directly leverages the Fill-Path mechanism to define a supervision signal aligned with fill-in formation.

## 3. Causal Structures for Fill-ins

We focus on sparse *symmetric* matrices $A \in \mathbb{R}^{n \times n}$ and study fill-in at the symbolic level. For nonsymmetric matrices, we first consider their structurally symmetrized form, typically represented by the sparsity pattern of $A + A^T$. With this convention, it is convenient to represent the symmetric sparsity pattern by an undirected graph $G(A) = (V, E)$, where $V = \{1, \ldots, n\}$ and $(i, j) \in E$ if and only if $a_{ij} \neq 0$ for $i \neq j$ (Harary, 1969; Rose, 1972). A permutation (ordering) $\pi : V \to \{1, \ldots, n\}$ specifies the elimination sequence, where a smaller $\pi(\cdot)$ indicates earlier elimination. According to the Fill-Path Theorem (Rose et al., 1976), the fill pattern of a sparse matrix is determined purely by the sparsity structure of $A$ and the ordering $\pi$, independent of numerical values. Here we utilize the causal structure to describe the causes of fill-ins and explore how to identify these structures.

### 3.1. Fill-in Formation

Sparse matrix factorization $A = LU$ is a fundamental step in solving sparse linear systems, as it represents the original matrix as the product of a lower triangular factor $L$ and an upper triangular factor $U$. Computationally, the factorization can be viewed as a sequence of elimination steps. At each step, a pivot row and pivot column are selected, the corresponding pivot variable is eliminated, and the remaining submatrix is updated. Fill-in refers to new nonzero positions created in the factors during elimination, even though those positions are zero in the original matrix. Each fill-in in the factored matrix $L + U$ is corresponding to a fill-edge in the graph. Symbolically, when a vertex $u$ is eliminated, all edges incident to $u$ are removed from the current graph. Meanwhile, all currently remaining neighbors of $u$ are connected into a clique, and any newly created edges are exactly the fill edges introduced at that step (Liu, 1990). After all vertices are eliminated, the collection of newly created edges characterizes the fill-in pattern, which corresponds to the

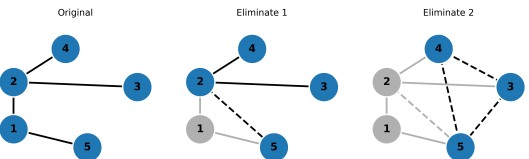

*Figure 1.* Symbolic elimination following the natural order. Eliminated vertices and incident edges are shown in gray, while newly created fill edges—introduced when the remaining neighbors of the eliminated vertex are connected into a clique—are drawn as dashed lines.

nonzero entries that appear in the resulting triangular factors $L$ and $U$, but are absent from the original matrix $A$.

To demonstrate the emergence of fill edges through successive clique completions, symbolic factorization steps are depicted in Figure 1. Each vertex elimination step proceeds by first removing the vertex and its incident edges, then creating possible fill edges among its neighbors to form a clique. Specifically, all pairs of its neighbor vertices are examined. For any pair that is not adjacent, a fill edge is required. For instance, eliminating vertex 1 requires the addition of fill edge $(2, 5)$ between vertices 2 and 5 to establish a clique in Figure 1. Similarly, eliminating vertex 2 completes a clique on the set $\{3, 4, 5\}$ by introducing three fill edges: $(3, 4)$, $(4, 5)$ and $(3, 5)$. The theoretical basis for this process is provided by Theorem 3.1.

### 3.2. Fill-Path and Its Inductive Reduction

The Fill-Path Theorem (Rose et al., 1976) establishes that a fill edge emerges if and only if there exists a path whose every interior vertex is eliminated earlier than both terminal vertices. However, direct enumeration of all such qualifying paths is computationally intractable in practice. To overcome this, we demonstrate that any fill-generating path of arbitrary length can be effectively reduced to a path of three vertices, where the middle vertex is eliminated before the two endpoints.

**Theorem 3.1** (Fill-Path Theorem (Rose et al., 1976)). *A fill edge between two distinct vertices $u$ and $v$ is created during elimination under $\pi$ if and only if there exists a path $P = (u = x_0, x_1, \ldots, x_{k-1}, x_k = v)$ in the original graph $G(A)$ such that every intermediate vertex on the path is eliminated before both endpoints, that is,*

$$\pi(x_i) < \min\{\pi(u), \pi(v)\}, \quad \text{for all } i = 1, \ldots, k-1.$$

Let $w \in \{x_i\}_{i=1}^{k-1}$ be the intermediate vertex on $P$ and that is eliminated last. Then $w$ splits the original path $P = u \sim v$ into two subpaths $u \sim w$ and $w \sim v$. Both subpaths $u \sim w$ and $w \sim v$ still satisfy the fill-in generation condition because $w$ is the last intermediate vertex on $P$ to be eliminated. For subpath $u \sim w$, all its intermediate vertices are elim-

inated earlier than $u$ and $w$. For subpath $w \sim v$, all its intermediate vertices are eliminated earlier than $w$ and $v$. Recursively, each subpath is partitioned into two subpaths according to the last eliminated intermediate vertex until the original path is collapsed into a collection of three-vertex paths, each satisfying that the intermediate vertex is eliminated earlier than the two terminals. Thus, we conclude that sampling fill-paths of length two, referred to as candidate triplets in Figure 2(b), is both sufficient and efficient for fill-in prediction, eliminating the need to enumerate all possible fill-paths.

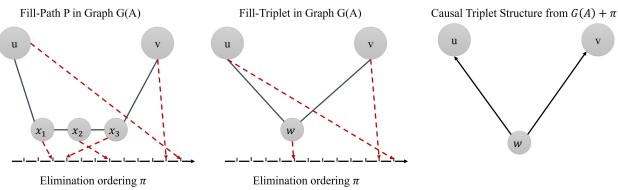

Fill-Path P in Graph G(A)   Fill-Triplet in Graph G(A)   Causal Triplet Structure from $G(A) + \pi$

Elimination ordering $\pi$        Elimination ordering $\pi$

*(a)* Fill-Path     *(b)* Candidate Triplet    *(c)* Causal Structure

*Figure 2.* Causal Structure for Fill-ins

### 3.3. Candidate Triplet as Causal Structure

We formalize the candidate triplet $u - w - v$ from the undirected graph $G(A)$ with intermediate vertex elimination first as a directed causal structure $u \leftarrow w \rightarrow v$ in Figure 2(c) (Spirtes & Glymour, 1991; Spirtes et al., 2000; Lagemann et al., 2023). The directed edge $w \rightarrow u$ means vertex $w$ is eliminated earlier than $u$, i.e. $\pi(w) < \pi(u)$. Similarly, $\pi(w) < \pi(v)$ is modeled as the direct edge $w \rightarrow v$. This specific structure is the cause of the fill-in at the position of $(\pi(u), \pi(v))$ in the factorized matrix.

Given a graph $G(A)$ and the elimination order $\pi$, the causal structure discovery process is as follows. For each nonadjacent vertex pair $(u, v) \notin E$, the possible causal structures can be enumerated via the intersection of their neighbors $\mathcal{N}_u \cap \mathcal{N}_v$. For each vertex $w$ in the intersection set, the candidate triplet $(u, w, v)$ is a causal structure if it satisfies $\pi(w) < \min(\pi(u), \pi(v))$. The time complexity of candidate triplet sampling process is about $\mathcal{O}(n^2)$ due to the sparsity of $G(A)$.

### 3.4. Causal Structure Learning for Fill-in Reduction

The Fill-Path condition in Theorem 3.1 states that fill-in between $u$ and $v$ can occur only when there exists a connecting path whose intermediate vertices are all eliminated before either endpoint. It is global since it quantifies over paths of arbitrary length. Section 3.2 has already presented an inductive reduction of this condition. Here, we provide a rigorous proof that, for symmetric sparsity, avoiding fill-in between the endpoints can be reduced to blocking the causal structures generated by candidate triplets.

For an ordering $\pi$, we associate each candidate triplet $(u, w, v)$ with $(u, w) \in E$, $(w, v) \in E$, and $(u, v) \notin E$ with the constraint

$$\pi(w) > \min\{\pi(u), \pi(v)\}, \tag{1}$$

meaning that the middle vertex $w$ is eliminated after at least one endpoint.

**Proposition 3.2** (Causal structure constraints are equivalent to global no-fill). *An ordering $\pi$ generates no fill edges if and only if Eq.(1) holds for every candidate triplet in $G(A)$, which equivalently means that $\pi$ blocks every triplet from forming the causal structure.*

*Proof.* (*Only if.*) If a candidate triplet $(u, w, v)$ violates Eq.(1), then $\pi(w) < \min\{\pi(u), \pi(v)\}$ and the length-two path $u$–$w$–$v$ satisfies Theorem 3.1; hence a fill edge between $u$ and $v$ must be created.

(*If.*) We prove the contrapositive. Suppose $\pi$ creates a fill edge between $u$ and $v$. By Theorem 3.1, there exists a $u$–$v$ path $P = (u = x_0, \ldots, x_k = v)$ whose intermediate vertices satisfy $\pi(x_i) < \min\{\pi(u), \pi(v)\}$. Choose such a path with the smallest possible length. Then for every $i$, $(x_{i-1}, x_{i+1}) \notin E$; otherwise the chord $(x_{i-1}, x_{i+1})$ would yield a strictly shorter path. Now let $x_j$ be the intermediate vertex on $P$ with the smallest $\pi(\cdot)$. By minimality, $\pi(x_j) < \pi(x_{j-1})$ and $\pi(x_j) < \pi(x_{j+1})$, so the candidate triplet $(x_{j-1}, x_j, x_{j+1})$ violates Eq.(1). Therefore, if all candidate triplets satisfy Eq.(1), no fill edge can be created. $\square$

This equivalence lets us replace global path supervision with a compact set of causal structure constraints instantiated by length-2 paths, which can be enforced efficiently while staying consistent with the fill-in formation mechanism.

## 4. Method

Through theoretical analysis above, we propose a complete **C**ausal **T**riplet **S**tructure identification framework for sparse matrix fill-in reduction, referred to as CTS. The whole framework is shown in Figure 3. It is mainly composed of vertex reordering network, candidate triplet sampling process and causal structure reasoning modules. For model optimization, we reduce the fill-ins by minimizing the number of causal triplets, which is approximate to increase the number of non-causal triplets by maximizing the likelihood function.

### 4.1. Vertex Reordering Network

For each sparse matrix $A \in \mathbb{R}^{n \times n}$, its sparsity pattern is represented as a graph $G = (V, E)$, where the number of vertices is $|V| = n$. Graph neural networks are naturally well-suited for encoding local neighbor structure into vertex

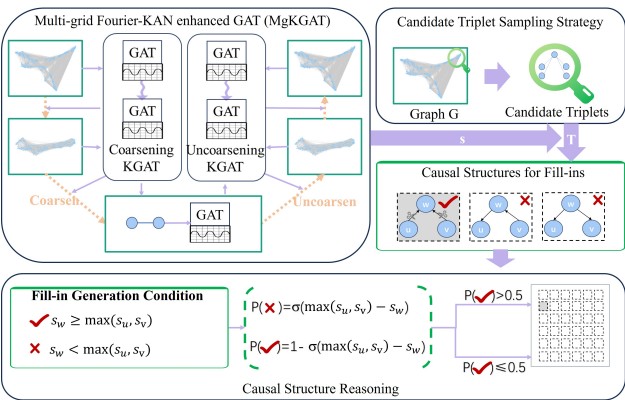

*Figure 3.* Causal Triplet Structure Inference Process

embeddings. To incorporate global structure information during initialization, spectral embedding of the graph Laplacian is introduced to capture the global position for each vertex. A graph attention network is then employed for vertex encoding, further enhanced by Fourier-KAN as the activation function. This KGAT network helps alleviate the gradient vanishing problem and enables the model to capture complex and long-range dependencies. Moreover, a multi-grid architecture is adopted to facilitate efficient structure information exchange between global and local graph levels. The overall framework of the proposed Multi-grid Fourier-KAN enhanced Graph ATtention network, i.e., MgKGAT, is illustrated in Figure 3.

### 4.1.1. Spectral Embedding for Initialization

The second eigenvector of the graph Laplacian is known as the Fiedler vector. It captures global connectivity and community structure while retaining sensitivity to local adjacency patterns. Consequently, the Fiedler vector provides a principled initialization for downstream vertex embedding in vertex reordering network. However, the exact computation of this second eigenvector is computationally expensive, typically scaling as $\mathcal{O}(n^3)$. To enable efficient initialization, we adopt a pre-existing spectral embedding module (Gatti et al., 2021) with graph neural network backbone and compute a low-dimensional approximation to the Fiedler vector of the graph Laplacian $L(G)$.

### 4.1.2. KAN enhanced Graph Attention Network

Taking this approximate global position of vertices as input, the proposed MgKGAT (Multi-grid KAN-enhanced Graph ATtention Network) predicts a scalar score for each vertex. Its basic unit, KGAT (Fourier-KAN enhanced Graph Attention Network), is to perform message passing over each graph level in the multi-grid computation process. Graph attention network is utilized to differentiate neighbor contributions for the irregular structure information with the attention mechanism. Fourier-KAN (Reinhardt et al., 2025) is adopted as activation for this graph attention network. The underlying reason is that Fourier-KAN introduces learnable frequency components that can capture complex and long-range dependencies in graph signals, while common activations are either piecewise linear or tend to saturate, suffering from vanishing gradients. Additionally, Fourier-KAN's spectral basis provides higher expressivity and smoother function approximation without the gradient limitations of fixed non-linearity.

For coarsening level $t$ ranging from 0 to $D-1$, the graph is denoted as $G^t = (V^t, E^t)$ and the corresponding vertex representation $x^t \in \mathbb{R}^{|V^t| \times d}$ is initialized from vertex embedding $\tilde{x}^{t-1} \in \mathbb{R}^{|V^t| \times d}$ at level $t-1$, where $d$ is the hidden dimension. Each coarsening Fourier-KAN enhanced Graph Attention (KGAT) layer proceeds by first aggregating structure information from neighbors as Eq.(2), and then nonlinearly mapping each hidden dimension of the aggregated information via Fourier basis like Eq.(3). Both $\omega_{j,i,k}^0$ and $\omega_{j,i,k}^1$ are learnable parameters of Fourier basis and $g_s$ is the grid size. $L$ layers of KGAT are performed over graph $G^t$.

$$h^t = \text{GATConv}_l^c(\tilde{x}^{t-1}, G^t) \tag{2}$$

$$x_j^t = \sum_{i=0}^{d-1} \sum_{k=0}^{g_s-1} \omega_{j,i,k}^0 cos(kh_{j,i}^t) + \omega_{j,i,k}^1 \sin(kh_{j,i}^t) \tag{3}$$

For uncoarsening level $t$ ranging from $D-1$ to 0, graph $G^t$ is refined from $G^{t+1}$. Initialized with vertex embedding $\hat{x}^{t+1} \in \mathbb{R}^{|V^t| \times d}$, each uncoarsening KGAT layer proceeds like coarsening KGAT layer in Eq.(4). FourierKAN($\cdot$)[1] is short for a dimension-wise Fourier-KAN nonlinearity like Eq.(3). $L$ uncoarsening layers of KGAT are performed over graph $G^t$.

$$x^t = \text{FourierKAN}\big(\text{GATConv}_l^{uc}(\hat{x}^{t-1}, G^t)\big) \tag{4}$$

### 4.1.3. Multi-grid Architecture

The proposed MgKGAT embeds $L$ coarsening KGAT layers before each graph clustering and $L$ uncoarsening KGAT layers after recovering each graph in the multi-grid pipeline. The initial spectral embedding $x \in \mathbb{R}^{|V| \times 1}$ is linearly transformed into higher-dimension vertex representations $x^0 \in \mathbb{R}^{|V| \times d}$, fed into MgKGAT method.

Starting from the original graph, at each level $t$ we first apply $L$ layers of KGAT by repeatedly using Eq.(3). We then cache the resulting pre-smoothed embeddings as a skip feature $x_{skip}^t$ and coarsen the graph via Graclus clustering followed by average pooling, which produces a smaller graph at level $t+1$ with embeddings $x^{t+1}$. After reaching

---

[1]https://github.com/byin-cwi/Toy-KAN

the coarsest level, we apply one additional KGAT update of the form Eq.(3) to inject global context.

During refinement, we traverse levels in reverse order. At each level $t$, we prolongate the coarse embedding back to the current vertex set through the recorded clustering map $P_t$, denoted by $\tilde{x}^t = P_t x^{t+1}$, and fuse it with the cached skip feature via $x^t \leftarrow \frac{1}{2}(\tilde{x}^t + x^t_{\text{skip}})$. After each such fusion, we apply $L$ uncoarsening KGAT layers using Eq.(4) to refine local details on the current graph. Finally, the refined embeddings at the finest level are fed into a linear projection followed by LeakyReLU to produce per-vertex scores $s$, i.e., $s = f_\theta(x)$, where $f_\theta$ denotes the proposed network with parameters $\theta$. The resulting scores $s$ are then used to induce the elimination ordering.

### 4.2. Causal Structure Reasoning

We realize the causal structure reasoning module as the matching between the candidate triplets $\mathcal{T}$ and reordering scores $s$ predicted from the reordering network mentioned above. Here we briefly introduce the candidate triplet sampling process in Alg.1. First, for each vertex in the graph $G(A)$, we build its neighbor set. Then, for each vertex $w$, we sample all possible vertex pairs $(u, v)$ from its neighbors and only keep those pairs for which there is no edge between $u$ and $v$ in $G(A)$.

Let $s_u$ denote the score predicted for vertex $u$. We induce the elimination ordering by sorting scores in descending order, so a larger score indicates earlier elimination. By Proposition 3.2, preventing fill along a candidate triplet $(u, w, v)$ is equivalent to requiring that the middle vertex $m$ is not eliminated before both endpoints. To align the ranking scores with constraints derived from the the proposition, this condition can be written as $s_w < \max(s_u, s_v)$, i.e., $w$ must score lower than at least one endpoint.

We define the triplet margin $z_{uwv} = \max(s_u, s_v) - s_w$, and interpret it probabilistically by mapping the margin to the probability that the constraint is satisfied:

$$p_{uwv} = \sigma(z_{uwv}) = \Pr(s_w < \max(s_u, s_v)), \quad (5)$$

where $\sigma(\cdot)$ denotes the sigmoid function. If $p_{uwv} < 0.5$, blocking $(u, w, v)$ fails and the triplet is classified as a causal structure for a new nonzero at the position $(\pi(u), \pi(v))$ of the reordered matrix $A'$ according to $s$ decreasingly.

### 4.3. Optimization Algorithm

Reducing fill-ins means maximizing the probability that the constraint is satisfied. Thus maximizing the likelihood function over the extracted candidate triplets $\mathcal{T}$ is equivalent to minimizing the negative log likelihood function as Eq.(6).

$$\mathcal{L} = \frac{1}{|\mathcal{T}|} \sum_{(u,w,v)\in\mathcal{T}} -\log \Pr(s_w < \max(s_u, s_v)) \quad (6)$$

**Algorithm 1** Candidate Triplet Sampling Process

---

**Require:** Given a graph $G(A) = (V_A, E_A)$
**Ensure:** candidate triplets $\mathcal{T}$
1: $\mathcal{T} = \{\}$
2: **for** $w \in V_A$ **do**
3: $\quad \mathcal{N}_w = \mathcal{N}_w \cup \{w\}$
4: **end for**
5: **for** $w \in V_A$ **do**
6: $\quad$ **if** $|\mathcal{N}_w| < 2$ **then**
7: $\quad\quad$ continue
8: $\quad$ **end if**
9: $\quad$ **for** $u, v \in \mathcal{N}_w \wedge (u, v) \notin E_A$ **do**
10: $\quad\quad \mathcal{T} = \mathcal{T} \cup \{(u, w, v)\}$
11: $\quad$ **end for**
12: **end for**
13: **return** $\mathcal{T}$

---

Since the likelihood function over each triplet is further reduced to $-\log \sigma(z) = \log(1 + e^{-z})$, this objective penalizes negative margins and drives $z_{uwv}$ to be large and positive, thereby promoting $s_w < \max(s_u, s_v)$ across candidate triplets. In practice, we implement the sigmoid function in a single, numerically stable operation. During training, for each input matrix, we enumerate all candidate triplets from the original graph and optimize the causal triplet loss in Eq.(6) to discourage the formation of causal triplet structures. The network parameters $\theta$ are updated via backpropagation using the Adam optimizer.

## 5. Experiments

We train our proposed framework on widely used benchmark datasets for sparse linear systems and conduct extensive evaluations on the SuiteSparse Matrix Collection (Davis & Hu, 2011), which comprises matrices arising from real-world engineering and scientific applications. Both fill-ins and causal structures are investigated in the experiments. The source code is publicly available at https://github.com/plumvvvv/CTS.

### 5.1. Experimental Setup

Our training data follows benchmark datasets that are widely used for sparse matrix problems (Gatti et al., 2021; Li et al., 2026; 2025). It consists of some matrices generated from Delaunay triangulations of various geometric shapes and finite-element discretizations, and other matrices sampled from the 2D/3D discretization category (2D3D) in the SuiteSparse Matrix Collection. Spectral embedding stage was trained with 5,000 matrices whose sizes range from 100 to 5,000. Vertex embedding stage was trained with 100 matrices of size from 200 to 500.

The test sets are from SuiteSparse Matrix Collection. For comprehensive evaluation, we sampled 69 matrices of size larger than 10,000 from diverse application domains, including Structural Problem (SP), Computational Fluid Dynamics (CFD), Chemical Process Simulation Problem (CPS), 2D/3D discretizations (2D3D), and others. For detailed

*Table 1.* Performances on a Test Matrix Set from different problems in SuiteSparse Matrix Collection.

*(a)* Fill-in Ratios

| | 2D3D | | CPS | | SP | | CFD | | others | | all | |
|---|---|---|---|---|---|---|---|---|---|---|---|---|
| | mean | std | mean | std | mean | std | mean | std | mean | std | mean | std |
| Natural | 730.59 | 1044.25 | 188.67 | 104.61 | 178.68 | 214.58 | 431.89 | 621.96 | 159.74 | 239.94 | 99.27 | 462.20 |
| AMD | 537.78 | 643.33 | 203.09 | 110.06 | 187.99 | 230.21 | 420.27 | 433.92 | 222.65 | 205.53 | 158.74 | 322.20 |
| ND | 83.64 | 74.73 | 166.26 | **69.19** | 60.13 | 130.42 | 76.25 | 71.45 | **62.18** | **78.15** | 50.12 | 92.73 |
| Spectral | 88.95 | 72.20 | 181.77 | 128.40 | 66.52 | 110.67 | 66.41 | 51.17 | 71.96 | 86.97 | 47.05 | 97.42 |
| UDNO | 85.69 | 82.51 | 177.36 | 92.07 | **48.64** | **91.76** | 58.18 | 55.07 | 66.10 | 89.77 | 47.06 | 91.60 |
| PMF | 81.01 | 72.38 | 173.87 | 75.90 | 57.09 | 126.04 | 56.35 | 53.40 | 66.92 | 90.86 | 45.92 | 96.09 |
| CTS | **80.02** | **66.16** | **160.43** | 70.74 | 52.17 | 95.17 | **54.59** | **50.27** | 64.90 | 88.98 | **45.05** | **85.54** |

*(b)* LU Factorization Time (second)

| | 2D3D | | CPS | | SP | | CFD | | others | | all | |
|---|---|---|---|---|---|---|---|---|---|---|---|---|
| | $t$(sec) | $\frac{t_{nat}}{t}$ | $t$(sec) | $\frac{t_{nat}}{t}$ | $t$(sec) | $\frac{t_{nat}}{t}$ | $t$(sec) | $\frac{t_{nat}}{t}$ | $t$(sec) | $\frac{t_{nat}}{t}$ | $t$(sec) | $\frac{t_{nat}}{t}$ |
| Natural | 1241.33 | - | 85.54 | - | 2118.40 | - | 670.27 | - | 57.41 | - | 688.18 | - |
| AMD | 942.77 | 1.73 | 120.57 | 0.73 | 391.05 | 2.81 | 496.74 | 1.89 | 418.69 | 42.12 | 438.68 | 18.82 |
| ND | 14.00 | 68.09 | 60.47 | 1.59 | 213.89 | 192.01 | 9.14 | 82.43 | 17.42 | **44.29** | 61.35 | 76.64 |
| Spectral | **9.44** | 169.38 | 84.79 | 1.36 | 52.06 | 108.51 | 6.19 | 167.02 | 28.64 | 28.13 | 35.52 | 75.41 |
| UDNO | 14.62 | 153.57 | 102.14 | 1.09 | **42.49** | 106.82 | 7.93 | 105.56 | 38.96 | 31.58 | 40.95 | 65.97 |
| PMF | 12.49 | 156.14 | 73.55 | 1.33 | 60.15 | **292.61** | 7.52 | 173.14 | 50.41 | 28.16 | 45.34 | **112.31** |
| CTS | 10.53 | **186.38** | **54.24** | **1.96** | 48.37 | 198.89 | **5.02** | **185.77** | 15.27 | 42.69 | **25.10** | 104.38 |

analysis, we selected all the matrices with size smaller than 1,000,000 from CPS and CFD in SuiteSparse Matrix Collection, and obtained 50 CPS matrices and 67 CFD matrices.

Three kinds of baselines are considered. Natural ordering means no reordering. Classical graph-theoretic methods are AMD, Nested Dissection (Metis), and a Fiedler-based spectral ordering. Deep learning methods include SE using an approximate Fiedler vector, UDNO promoting diagonal concentration, and PFM reducing fill-in via differentiable approximate factorization. All the deep learning methods are trained on the same training set. All the baselines and CTS are evaluated on three test benchmark datasets. Hyperparameters of SE, PFM and UDNO are set according to the original papers.

Hyperparameters of CTS are set as follows. Learning rates are chosen from 0.1 to 1e-5. The number of KGAT layers $L$ is selected from 1 to 5. Grid size $g_s$ in FourierKAN is selected from 3,...,100,300. The dimension of hidden layers is selected from 8, 16, 32, 64. All these hyperparameters are chosen by minimizing the fill-in ratio on the training set. Their corresponding values are 0.002, 2, 3, 16 respectively. The computation iteration number $D$ in multi-grid method is dependent on the specific graph, usually no more than 20 in practice. More details can be found in the source code.

Two metrics are used to evaluate the effectiveness of the proposed method. Fill-in ratio (FIR) measures the relative growth in nonzeros caused by factorization after reordering and thus reflects the increase in memory and computation costs:

$$\text{FIR} = \frac{\text{nnz}(L + U - I) - \text{nnz}(A)}{\text{nnz}(A)}. \qquad (7)$$

Lower FIR indicates fewer fill-ins. LU factorization time after reordering reflects the practical acceleration achieved in the factorization stage using a Dell Precision 3680 Workstation (a tower workstation from Dell Technologies). For

each test matrix, we apply each ordering method to obtain a permutation and reorder the matrix accordingly. We then factorize the reordered matrix using SuperLU.

### 5.2. Performance Analysis

We report the mean and its standard deviation of fill-in ratio and LU factorization time per test problem group for each reordering method in Table 1. Fewer fill-ins mean less LU factorization time. The proposed method Causal Triplet Structure Learning (CTS) consistently achieves the best fill-ins across all program groups in terms of the mean fill-in ratio. The number of fill-ins is 13× lower than that achieved by the state-of-the-art deep learning method PFM and 6× lower than classical graph-theoretic method ND on CPS matrix subset. CTS achieves the second best on SP and other subsets. This suggests the generalization ability of CTS is limited on some problems, which will be explored in the following subsection.

In terms of LU factorization time in Table 1, CTS achieves 12× speedup compared with PFM on CFD subset and 16× speedup compared with the best baseline, Spectral Ordering, on the same subset. LU factorization is highly dependent on the hardware and running environment. Though the LU factorization time comparison results do not strictly align with fill-in ratio results, CTS tends to be a better choice for most cases. It's worth noting that state-of-the-art results are usually achieved by deep learning methods on all the subsets.

The performance of CTS varies across different types of matrices. Compared with CFD, the smaller gain of CTS on SP is largely attributed to matrix structure. Many SP matrices exhibit banded or near-diagonal sparsity patterns. This is particularly well aligned with UDNO, whose design explicitly encourages nonzeros to concentrate more tightly around the diagonal. In contrast, CTS is derived

from the mechanism of fill generation itself and is intended to serve as a more general framework applicable across matrix types. Although CTS does not outperform UDNO on SP, it remains competitive on this matrix type and still outperforms the other baselines, while achieving the best overall performance across the entire benchmark suite. A more detailed analysis across applications will be presented in the subsequent subsection.

*Table 2.* Ablation Study of CTS.

| | | | CPS | CFD | CPS+CFD |
|---|---|---|---|---|---|
| SE (spectral embedding) | | | 182.90 | 61.00 | 118.74 |
| SE | MgSAGE | $l_{\text{CTS}}$ | 174.02 | 54.81 | 111.28 |
| | MgGAT | | 164.82 | 59.47 | 109.37 |
| | KGAT | | 167.41 | **52.83** | **107.10** |
| | GUnet | | 188.52 | 142.29 | 164.19 |
| | MgKGAT | $l_{\text{PFM}}$ | 185.79 | 58.93 | 119.02 |
| CTS(SE+MgKGAT+$l_{\text{CTS}}$) | | | **160.43** | 54.59 | 107.51 |

## 5.3. Ablation Study

CTS is mainly composed of Reordering Network MgKGAT, Causal Triplet Loss, and Causal Structure Reasoning modules. We investigate roles of the former two modules in Table 2. MgKGAT utilizes a multi-grid architecture with KGAT for vertex embedding per level, i.e. FourierKAN as activation function for Graph Attention Network (GAT).

With fixed causal triplet loss function $l_{\text{CTS}}$, two multi-grid architectures with different graph networks, i.e. MgGAT and MgSAGE are used for reordering. KGAT and GUnet are without multi-grid architectures. Comparison results show that multi-grid architectures are more robust to the choice of graph network. However, the graph network design is key to the performance without multi-grid. The carefully designed KGAT is even better than MgKGAT on regular CFD matrices, which are simpler than irregular CPS matrices. Given the reordering network MgKGAT, the proposed causal triplet loss function also plays a significant role in the fill-in ratio reduction.

## 5.4. Effectiveness of Causal Structure Reasoning

Here the causal structure identification problem is solved by a binary classification task over sampled candidate triplets. For a given candidate triplet $(u, w, v)$ in a graph $G(A)$, the predicted probability that it constitutes a causal structure is defined as $1 - p_{uwv}$, where $p_{uwv}$ is defined in Eq.(5). If this probability exceeds 0.5, candidate triplet $(u, w, v)$ is identified as a causal structure for the fill-in at position $(\hat{\pi}(u), \hat{\pi}(v))$ in the factorized matrix $L + U$.

The proposed causal structure reasoning module takes the sampled candidate triplets and reordering score as input from the sampling strategy and reordering network respectively. To assess the validity of this module, we report the alignment between the predicted fill-in pattern and the actual fill-in pattern using the IoU and Dice metrics (Jaccard, 1901;

Dice, 1945) on the CPS matrix subset of the mixed dataset, as shown in Figure 4. In this evaluation, PFM denotes the combination of our candidate triplet sampling method with the reordering network from PFM.

| | Derived Fill-in Pattern | | | | Fill-in Ratio | |
|---|---|---|---|---|---|---|
| | IoU(%) | | Dice(%) | | | |
| | PFM | CTS | PFM | CTS | PFM | CTS |
| lhr10 | 2.75 | 2.80 | 5.35 | 5.43 | 112.51 | 111.15 |
| lhr11 | 2.42 | 3.04 | 4.73 | 5.90 | 129.05 | 104.68 |
| bayer10 | 0.78 | 0.74 | 1.54 | 1.47 | 355.79 | 322.61 |
| lhr10c | 2.60 | 2.90 | 5.06 | 5.63 | 108.49 | 97.64 |
| lhr14c | 2.22 | 2.23 | 4.35 | 4.37 | 170.91 | 163.72 |
| lhr17 | 1.60 | 2.14 | 3.14 | 4.19 | 176.57 | 170.82 |
| lhr17c | 1.82 | 2.00 | 3.57 | 3.90 | 187.29 | 164.46 |
| lhr34c | 1.46 | 1.77 | 2.88 | 3.47 | 197.24 | 199.38 |
| lhr11c | 2.84 | 3.18 | 5.52 | 6.16 | 127.01 | 109.44 |

*(a)* Accuracy of fill-in patterns derived from learning methods with and without causal structure modeling, i.e., CTS and PFM. All the matrices can be indexed by their names in the left column via `https://sparse.tamu.edu/`.

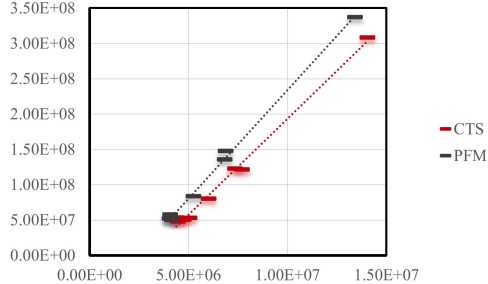

*(b)* No. of Discovered Causal Structures ($x$-axis) vs. No. of New Nonzeros from LU Factorization ($y$-axis). Correlation coefficients of both CTS and PFM are 1.0.

*Figure 4.* Effectiveness of Causal Structure Learning on CPS.

These identified causal structures are directly used to predict the fill-in pattern as a binary matrix without de facto factorization. The ground truth fill-in pattern is $L + U$'s binary form from the LU factorization of the reordered matrix. The overlap in terms of IoU and Dice between fill-in patterns shows the effectiveness of the causal structure reasoning. Without explicit causal structure modeling, both IoU and Dice of PFM are lower than CTS. The fill-in pattern overlap not only measures the fill-in amount, but also requires precise matching of fill-in locations. In addition to this stringent metric, we also summarize the fill-in pattern as the nonzero number and briefly compare the nonzero number correlation between fill-in patterns in Figure 4(b). As expected, the predicted fill-in pattern correlates with the ground truth. We therefore use causal structures for optimization.

## 5.5. Generalization to Sparse Matrices from Different Problems

CTS is trained on a mixed matrix dataset, including a matrix subset from 2D/3D problem in SuiteSparse Matrix Collection. As shown in Table 1, CTS achieves larger fill-in ratio

reductions on CPS and CFD than on the other subsets. Here we further explore the generalization performance of CTS and evaluate its learned model on matrices from both CPS and CFD problems.

**Chemical Process Simulation Problem (CPS).** CPS matrices are usually Jacobian matrices of steady-state or dynamic process system models, describing component, energy, and momentum balances. They are usually of relatively small scale and very sparse, with irregular structures. We select all the matrices in the Chemical Process Simulation category from SuiteSparse Matrix Collection with the matrix size (row number) less than $1,000,000$, and obtain 50 matrices after removing those that cannot be factorized. Both matrix size and fill-in ratio vary widely, and we report several statistics listed in Figure 5(b) to help obtain the whole picture of matrices. In the violin plot, CTS yields lower fill-in ratios than the other three methods. Even for a small matrix b1_ss, CTS captures the causal structures more precisely than PFM and obtains the best ratio as ND. The result suggests fill-in ratios of CTS are consistently lower than other methods in the figure, even on matrices from the unseen problems.

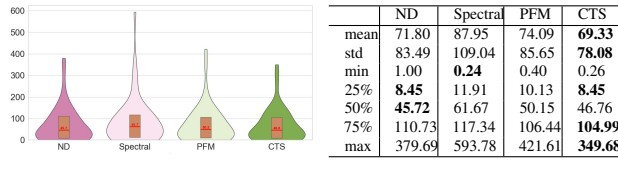

| | ND | Spectral | PFM | CTS |
|---|---|---|---|---|
| mean | 71.80 | 87.95 | 74.09 | **69.33** |
| std | 83.49 | 109.04 | 85.65 | **78.08** |
| min | 1.00 | **0.24** | 0.40 | 0.26 |
| 25% | **8.45** | 11.91 | 10.13 | **8.45** |
| 50% | **45.72** | 61.67 | 50.15 | 46.76 |
| 75% | 110.73 | 117.34 | 106.44 | **104.99** |
| max | 379.69 | 593.78 | 421.61 | **349.68** |

*(a)* Fill-in Ratio Distributions.  *(b)* Statistics.

*Figure 5.* Fill-in Ratios over More Matrices from CPS.

**Computational Fluid Dynamics Problem (CFD).** CFD matrices are often coefficient matrices from the discretization of fluid governing equations like Navier-Stokes. They are extremely large in scale and highly sparse, exhibiting a regular block structure. We select all the matrices in the Computational Fluid Dynamics category from SuiteSparse Matrix Collection with matrix size less than $1,000,000$, and obtain 67 matrices by removing those that cannot be factorized. Similar setting is mentioned above. Obviously, the fill-in ratio distribution of CTS is lower and fatter in Figure 6(a). This comparison result is corresponding to statistics in the right table. The comparison result between PFM and CTS also suggests the superiority of causal structure learning. Generally, it achieves robust and consistently better performance for CFD matrices.

**5.6. Scalability to Larger Sparse Matrices**

The test matrix size, row or column number, in the mixed set is more than $10,000$ while the training matrix size ranging from 200 to 500. To evaluate the scalability of the proposed method, we sample 7 groups from the mixed set according to their matrix sizes. Due to its sparsity, the matrix size

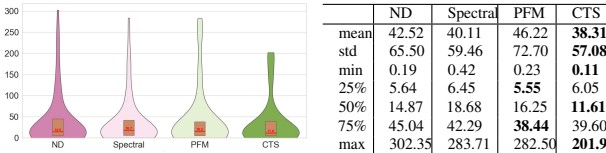

| | ND | Spectral | PFM | CTS |
|---|---|---|---|---|
| mean | 42.52 | 40.11 | 46.22 | **38.31** |
| std | 65.50 | 59.46 | 72.70 | **57.08** |
| min | 0.19 | 0.42 | 0.23 | **0.11** |
| 25% | 5.64 | 6.45 | **5.55** | 6.05 |
| 50% | 14.87 | 18.68 | 16.25 | **11.61** |
| 75% | 45.04 | 42.29 | **38.44** | 39.60 |
| max | 302.35 | 283.71 | 282.50 | **201.95** |

*(a)* Fill-in Ratio Distribution.  *(b)* Statistics.

*Figure 6.* Fill-in Ratio Distribution over More Matrices from CFD

in this experiment is the number of nonzeros in the matrix. Each group contains matrices with matrix size less than a certain value from 100 thousand to 15 million. Each point in Figure 7 is computed as the mean value of the group.

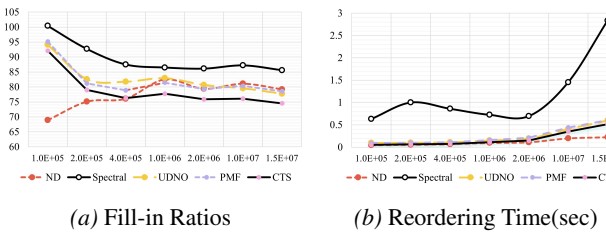

*(a)* Fill-in Ratios  *(b)* Reordering Time(sec)

*Figure 7.* Fill-in Ratio and Reordering Time Variation with the increase of Matrix Size on the mixed test Set.

According to Figure 7(a), the fill-in ratio like ND shows a sharp initial increase before 1 million, and then keeps stable. Other fill-in ratios from spectral ordering method and deep learning based methods decline a little before 1 million and then nearly stay steady. The result indicate all those deep learning methods behave similarly to those graph-theoretic methods and shows better scalability to large matrices on this mixed test set regardless of their training matrix sizes. The reordering time of spectral ordering method changes dramatically with the increase of matrix size while others are not. The substantial runtime variability of graph-theoretic methods like spectral ordering method across matrices motivates our adoption of deep learning techniques.

## 6. Conclusion and Limitations

Fill-in can greatly increase the time and memory costs of sparse direct solvers, while reordering can mitigate it. Building on the Fill-Path Theorem, we show that blocking causal triplet structures suffices to prevent fill generation, and propose CTS, which uses a multigrid KAN-enhanced GAT with a causal triplet loss to learn reordering scores. Experiments on SuiteSparse benchmarks demonstrate state-of-the-art fill-in reduction and consistent LU speedups.

Despite its strong performance, CTS has limitations. Its gains vary across matrix families for reasons not yet fully understood, and our objective does not explicitly model the impact of reordering on parallelism in parallel solvers.

## Acknowledgements

This research was supported in part by the National Key R&D Program of China under Grant No. 2021YFB0300203 and the National Natural Science Foundation of China under Grant No. 12471348.

## Impact Statement

This paper presents work whose goal is to advance machine learning for sparse numerical linear algebra. By reducing fill-in during sparse matrix factorization, our method can lower computational and memory costs, potentially improving efficiency and reducing energy consumption in scientific computing workflows. We do not identify any societal consequences that require specific additional discussion.

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
