# OpenReview forum: "Causal Structure Learning for Sparse Matrix Fill-in Reduction"
_ICML.cc/2026/Conference — ICML 2026 regular_

### Official Review · Reviewer_LZtc · 2026-03-11

**Soundness:** 3
**Presentation:** 3
**Significance:** 2
**Originality:** 2
**Overall Recommendation:** 4
**Confidence:** 3

**Summary:**

The paper tackles the fundamentally important and NP-hard problem of minimizing fill-in during the LU factorization of sparse matrices. While recent deep learning approaches have shown empirical success, they often lack interpretability regarding the actual mechanism of fill-in generation. To bridge this gap, the authors propose Causal Triplet Structure Learning (CTS), a novel framework grounded in the Fill-Path Theorem. The method cleverly reduces arbitrary-length fill-paths to length-two candidate triplets and formulates the minimization problem as identifying and blocking these causal structures. Architecturally, the authors employ a multigrid-style Graph Attention Network with KAN activations (MgKGAT) to learn vertex embeddings, supervised by a causal triplet loss. Empirical evaluations on the SuiteSparse Matrix Collection demonstrate that CTS can achieve up to a 6x reduction in fill-in and a 12x speedup in factorization time compared to state-of-the-art deep learning methods on specific subsets like CPS and CFD matrices.

**Compliance With Llm Reviewing Policy:**

Affirmed.

**Final Justification:**

This paper demonstrates high practical value. However, I find that the methodological contribution is limited, as the approach mainly consists of a combination of existing methods rather than introducing a fundamentally new idea. The authors’ rebuttal did not sufficiently address this concern.

**Key Questions For Authors:**

1. Regarding MgKGAT and KAN Activations: What is the fundamental motivation for using Fourier-KAN activations in this specific causal structure learning task? Could you provide an ablation study or theoretical intuition showing why standard MLPs or ReLU/GELU activations fail or underperform in capturing these triplet constraints compared to KANs?
2. Could you clarify what constitutes the core methodological innovation of your network design beyond the concatenation of existing modules (Fiedler initialization, Multigrid, GAT, KAN, BCE loss)? Are there unique, demonstrable synergistic effects between these specific components that justify this exact pipeline, or could a much simpler, unified architecture achieve similar results when supervised by your causal triplet loss?

**Limitations:**

Yes.

**Strengths And Weaknesses:**

- Strengths:
    - The evaluation is conducted on a widely recognized and diverse benchmark, the SuiteSparse Matrix Collection. The authors evaluate both the proxy metric (fill-in ratio) and the ultimate practical metric (LU factorization time), providing a realistic view of the method's utility.
    - The theoretical reduction of arbitrary-length fill-paths into candidate triplets is well-explained, and Proposition 3.2 effectively solidifies the justification for the proposed loss function.

- Weaknesses:
    - Clarity or presentation issues: The paper designs a multigrid-style GAT with Fourier-KAN activations (MgKGAT), but fails to provide a compelling, task-specific motivation for this architectural choice. It remains unclear why KANs are theoretically or empirically better suited for learning causal triplet structures compared to standard MLPs or traditional activation functions, leaving the reader confused as to whether this is a core methodological necessity or merely an orthogonal trick.
    - While the theoretical reduction of fill-paths is insightful, the proposed network architecture feels like an incremental amalgamation of existing techniques. The pipeline simply stacks well-known modules—spectral embeddings , a standard multigrid framework , GATs , Fourier-KAN activations , and a standard Binary Cross-Entropy (BCE) loss. The paper lacks a rigorous justification for why this specific, complex combination of trendy components is necessary, making the technical contribution appear more like an engineered pipeline than a fundamental architectural breakthrough.

---

> ### Author Rebuttal · Authors · 2026-03-31
>
> We thank the reviewer for these thoughtful questions.
>
> **Q1. On the motivation for using Fourier-KAN.**
>
> Our motivation for using Fourier-KAN is not to introduce an architecture-level trick, but to better match the nature of this task. The model is learning a node scoring function shaped jointly by local triplet constraints and global graph structure. This is not a standard smooth regression target: small structural changes in the graph may alter triplet violations and relative node rankings, which can in turn change whether fill is generated. As a result, the target function can be highly non-smooth and involve structural patterns at multiple scales.
> From this perspective, Fourier-KAN offers a useful inductive bias. Its Fourier basis functions make it more suitable for representing rapidly changing, oscillatory, and mixed-scale patterns than standard pointwise activations. This intuition is also supported by our ablation results in Table 2. In particular,SE+MgGAT+$l_{\mathrm{CTS}}$keeps the CTS framework unchanged and only replaces the Fourier-KAN activation with ReLU. Compared with this variant, the full CTS model consistently achieves lower fill-in ratios.
>
> **Q2. On the design rationale of the proposed framework.**
>
> We agree with the reviewer that the main contribution of the paper is not a brand-new neural network module. Rather, the core contribution is a learning framework derived from the mechanism of fill generation itself. Starting from the Fill-Path Theorem, we show that fill-generating paths of arbitrary length can be reduced to local triplet constraints, and we then use this reduction to construct the causal triplet loss. This makes it possible to optimize sparse matrix reordering directly around the source of fill generation, rather than through a more indirect surrogate.
> Under this view, MgKGAT is better understood as a task-matched instantiation of the framework than as a standalone architectural breakthrough. Its components are included for different reasons: spectral initialization provides a global ordering prior; the multigrid structure captures cross-scale dependencies; GAT models adaptive local interactions; Fourier-KAN helps represent the non-smooth ranking function induced by global structure and local triplet constraints; and the BCE-style objective matches the binary nature of triplet violation events.
> To make this clearer, we reorganized Table 2 into pairwise comparisons. The revised ablation results show that nearly all components contribute positively to the final performance. The only less consistent effect comes from the multigrid structure, whose benefit appears to depend on the matrix family; for example, on the CFD subset, the non-multigrid variant performs slightly better. Overall, these results suggest that the proposed method is not simply a loose combination of existing components, but a structured design in which each module plays a specific role in fill-aware reordering.
>
>
> | Group | Compared models | CPS | CFD | CPS+CFD |
> |---|---|---:|---:|---:|
> | Attention aggregation | SE+MgSAGE+$l_{\mathrm{CTS}}$ | 174.02 | **54.81** | 111.28 |
> |  | SE+MgGAT+$l_{\mathrm{CTS}}$ | **164.82** | 59.47 | **109.37** |
> | KAN activations | SE+MgGAT+$l_{\mathrm{CTS}}$ | 164.82 | 59.47 | 109.37 |
> |  | SE+MgKGAT+$l_{\mathrm{CTS}}$ | **160.43** | **54.59** | **107.51** |
> | Multigrid structure | SE+KGAT+$l_{\mathrm{CTS}}$ | 167.41 | **52.83** | **107.10** |
> |  | SE+MgKGAT+$l_{\mathrm{CTS}}$ | **160.43** | 54.59 | 107.51 |
> | Loss function | SE+MgKGAT+$l_{\mathrm{PFM}}$ | 185.79 | 58.93 | 119.02 |
> |  | SE+MgKGAT+$l_{\mathrm{CTS}}$ | **160.43** | **54.59** | **107.51** |
> | Overall framework | SE | 182.90 | 61.00 | 118.74 |
> |  | CTS (SE+MgKGAT+$l_{\mathrm{CTS}}$) | **160.43** | **54.59** | **107.51** |

---

> > ### Author Rebuttal · Reviewer_LZtc · 2026-04-01
> >
> > I thank the authors for their comprehensive response. While the clarifications are thorough, my assessment regarding the originality of this work remains unchanged. It appears that the methodology is more of a combination of existing techniques rather than a novel conceptual contribution. Nonetheless, I recognize its engineering value and practical utility. Therefore, I have decided to maintain my current recommendation.

---

> > > ### Author Response · Authors · 2026-04-03
> > >
> > > Thank you for your time and for carefully considering our rebuttal. We appreciate your thoughtful feedback.

---

### Official Review · Reviewer_JMVP · 2026-03-11

**Soundness:** 3
**Presentation:** 3
**Significance:** 3
**Originality:** 2
**Overall Recommendation:** 3
**Confidence:** 4

**Summary:**

This paper studies sparse matrix reordering for fill-in reduction. The main idea is to reduce global fill-generating paths to local candidate triplets, then train a neural reordering model (CTS) with a triplet-based loss so the middle node in a triplet is not eliminated before both endpoints. The method combines this loss with a multigrid GAT/KAN backbone and reports strong results on SuiteSparse benchmarks.

**Compliance With Llm Reviewing Policy:**

Affirmed.

**Key Questions For Authors:**

- The paper reports especially strong improvements on CPS and CFD, but also notes that gains vary and generalization is limited on some subsets. Do you have a structural explanation for when CTS should help most?
- Since fill-in reduction is a central motivation of the paper, I was surprised that the timing results do not always align closely with fill-in ratio. Can the authors explain this discrepancy more concretely? Method like ND, achieving low fill-in ration but suffering from high factorization time. Can the authors explain this?

**Limitations:**

yes

**Strengths And Weaknesses:**

Strengths:
- The paper connects the Fill-Path theorem to a local triplet condition, then turns that condition into a trainable loss. That gives the method a cleaner motivation than a generic learned surrogate.
- The paper does more than a headline table: it includes ablations, fill-pattern analysis, and scalability experiments, which help support the method beyond raw benchmark numbers.

Weaknesses:
- Novelty is not fully convincing. The triplet reduction seems closely related to classical sparse elimination / perfect-elimination characterizations, so the paper should distinguish more clearly what is actually new.
- The “causal” framing somewhat overstated: the paper identifies a structural condition for fill generation, but this is not causal inference in the usual machine learning sense.

---

> ### Author Rebuttal · Authors · 2026-03-31
>
> To Reviewer JMVP:
>
> We thank the reviewer for the thoughtful comments and helpful questions.
>
> **W1. On the relation to perfect-elimination characterizations and our novelty.**
> We agree that perfect elimination is a classical concept in sparse elimination theory. A perfect elimination ordering leads to no fill-ins. However, such an ordering does not exist for most sparse matrices in practice. Alternatively, the ultimate goal of sparse matrix reordering is to produce an ordering that induces as little fill as possible, ideally approaching perfect elimination whenever the matrix structure permits. Here we attempt to learn a reordering network to predict an elimination ordering with fewer fill-ins.
>
> The *Fill-Path Theorem* bridges the gap between elimination ordering and fill-ins. However, it requires enumerating global paths of all possible lengths to determine the fill-in existence for each matrix entry, which is not suitable for optimization. We reduce this global combinatorial condition into a local triplet form. It is easily converted to triplet constraints on the network predicted ordering for fill-in reduction. Our major contribution lies in the theoretical reduction to local triplet form and directly utilizing it as loss function for fill-in reduction. Meanwhile, these triplets are candidate causes for fill-ins.
>
> **W2. On the use of the term “causal”.**
> We agree with the reviewer that our use of *causal* is not intended in the sense of causal inference in machine learning. Rather, it refers to the structural cause-and-effect relationship between elimination order and fill generation: for a candidate triplet, eliminating the middle node before both endpoints causes a new fill edge to appear between the endpoints. Our intention was to emphasize this structural causality in the elimination process, not to claim a contribution in causal inference. We will revise the wording in the final version to make this distinction explicit and avoid possible overstatement.
>
> **Q1. On when CTS helps most.**
> CTS is designed from the underlying principle of fill generation and is therefore broadly applicable. Empirically, it is effective across all tested matrix groups and achieves the best fill-in ratio on most of them. The main exception is the SP category, where UDNO achieves a slightly lower fill-in ratio. SP matrices exhibit banded structure or strong concentration of nonzeros near the diagonal. This matches the inductive bias of UDNO, which explicitly encourages diagonal concentration and bandwidth reduction.
> In contrast, CTS is designed to suppress fill generation more directly through elimination-order constraints, rather than to explicitly promote diagonal concentration. We believe this explains why UDNO is particularly competitive on SP, while CTS performs best on the other matrix categories.
>
> **Q2. On why fill-in ratio and factorization time are not always perfectly aligned.**
> We agree that fill-in is a central motivation of our method and an important determinant of factorization time, but it is not the only one.
> Besides, wall-clock factorization time also depends on the computation organization and memory access by the solver, such as supernode amalgamation, pivot threshold, data locality and so on.
> Some orderings yield irregular supernode structures, where fewer but more scattered fill entries lead to less favorable dense updates, poorer locality, higher cache miss rates, and reduced kernel efficiency, so the actual runtime can still be longer even with less fill.
> As a result, the actual factorization time may remain relatively high. This also helps explain why methods such as ND can sometimes achieve low fill-in ratios but still exhibit higher factorization time: fill-in is a strong but incomplete proxy for runtime efficiency.
>
> We will incorporate these clarifications in the final version.

---

> > ### Author Rebuttal · Reviewer_JMVP · 2026-04-04
> >
> > Thank you for the rebuttal. While some concerns were clarified, novelty remains my main concern, and I will maintain my score.

---

> > > ### Author Response · Authors · 2026-04-07
> > >
> > > Thank you for your time and for carefully considering our rebuttal. We appreciate your thoughtful feedback.

---

### Official Review · Reviewer_L5ce · 2026-03-13

**Soundness:** 3
**Presentation:** 3
**Significance:** 4
**Originality:** 3
**Overall Recommendation:** 4
**Confidence:** 2

**Summary:**

The manuscript presents a framework for sparse matrix reordering termed Causal Triplet Structure (CTS) learning. The method utilizes a technical reduction of the Fill-Path Theorem to define local, three-vertex constraints that govern fill-in generation during LU factorization. The architectural implementation, MgKGAT, incorporates a multigrid Graph Attention Network featuring Fourier-based Kolmogorov-Arnold Network (KAN) activations to capture high-frequency structural signals. This system is optimized via a differentiable Causal Triplet Loss that penalizes score configurations failing to block fill-in according to the inductive triplet condition, thereby transforming a discrete combinatorial problem into a continuous probabilistic task.

Empirical evaluation on the SuiteSparse Matrix Collection focuses on Chemical Process Simulation (CPS) and Computational Fluid Dynamics (CFD) matrices. The authors report up to a 6x reduction in fill-in ratios and a corresponding 12x acceleration in LU factorization time compared to established deep learning baselines like PFM and UDNO. The framework is designed to generalize from small training instances to matrices with over 1,000,000 rows by leveraging the local nature of the triplet constraints.

**Compliance With Llm Reviewing Policy:**

Affirmed.

**Key Questions For Authors:**

Does the reordering time reported in the experiments include the candidate triplet sampling phase, and how does the O(N2) complexity of this step affect scalability on matrices with dense rows or hub nodes?
The manuscript evaluates performance using a sequential solver, but how does the CTS reordering affect the elimination tree height and subsequent parallelism in multi-threaded solvers like MUMPS or PARDISO?
Is the reported generalization from small to large matrices driven primarily by the local nature of the triplet loss or by the hierarchical features learned by the multigrid backbone?
What specific structural artifacts or performance benefits led to the selection of a Fourier basis for the KAN layers instead of the standard B-splines used in original Kolmogorov-Arnold implementations?
How does the causal triplet loss handle cases where multiple intermediate neighbors exist between a non-adjacent pair, and does it account for the combined probability of fill-in from multiple paths?

**Limitations:**

The authors document that performance improvements are domain-dependent and that the current objective does not explicitly model parallel solver requirements. However, the manuscript lacks a detailed discussion of the training-to-inference cost ratio, which is relevant given the high data requirements for neural reorderers. There is also no analysis of how reordering for fill-in reduction might negatively impact numerical stability or pivoting requirements in ill-conditioned systems. To improve the impact statement, the authors should consider highlighting the potential for reduced energy consumption in large-scale simulations as a positive societal outcome of more efficient sparse solvers.

**Strengths And Weaknesses:**

The primary technical strength lies in the formal proof of Proposition 3.2, which establishes an equivalence between global no-fill conditions and local triplet blocking. This allows for a differentiable optimization target that avoids the computational expense of symbolic or numeric factorization during the training loop. The integration of a multigrid backbone with Fourier-KAN layers is a novel architectural response to the spectral bias often found in traditional Graph Neural Networks applied to scientific computing. The performance gains in the CPS and CFD domains are substantial, demonstrating that the method effectively captures specific sparsity structures that classical heuristics like AMD often miss.

A significant weakness is the O(N^2) computational overhead of the candidate triplet sampling process, which is not clearly accounted for in the reported reordering time comparisons. In matrices with high-degree nodes, this sampling phase could dominate the total runtime, potentially negating the benefits of the learned ordering. Additionally, the evaluation utilizes a sequential SuperLU solver, which fails to assess the reordering's impact on elimination tree height-a critical metric for parallel efficiency in high-performance computing environments. The generalization of the method is also inconsistent, as gains in Structural Problems (SP) are significantly lower than in fluid dynamics, suggesting the local triplet assumption may not hold for all graph topologies.

---

> ### Author Rebuttal · Authors · 2026-03-31
>
> We thank the reviewer for the careful reading and constructive feedback.
>
> **1. On candidate triplet sampling and reported reordering time.**
> Candidate triplet sampling is used only during training to instantiate the CTS loss, and is not needed at inference time. Therefore, it does not affect the reported reordering time. At test time, CTS directly predicts node scores and sorts nodes in a descending order based on derived scores. We will make this point explicit in the final version.
>
> **2. On elimination tree height and parallelism.**
> We agree that fill-in and elimination tree height are both important for sparse reordering, but they correspond to different objectives. CTS is designed specifically for fill-in reduction in the sequential LU setting, and does not explicitly optimize elimination tree height or downstream parallelism. Jointly optimizing fill-in and parallel structure is an important direction for future work. In follow-up work, we plan to incorporate graph partitioning and related structural objectives into the ordering process to better support parallel solvers.
>
> **3. On the smaller gains on SP matrices.**
> The smaller gain of CTS on SP than on CFD is largely explained by matrix structure. Many SP matrices already have limited room for fill-in reduction under natural ordering and exhibit banded or near-diagonal sparsity patterns. This is particularly well aligned with UDNO, whose design explicitly encourages nonzeros to concentrate more tightly around the diagonal. By contrast, CTS is derived from the mechanism of fill generation itself and is intended as a more general framework across matrix types. Although CTS does not outperform UDNO on SP, it remains competitive there and still outperforms the other baselines, while achieving the best overall performance across the full benchmark suite.
>
> **4. On the contribution of the triplet objective versus the multigrid backbone.**
> Our ablation results suggest that both components are useful, but the triplet objective plays the larger role. On CPS, replacing the CTS loss with the PFM loss while keeping the architecture fixed ($SE+\mathrm{MgKGAT}+l_{\mathrm{PFM}}$) increases the fill-in ratio by 25.36, whereas removing the multigrid structure while keeping the CTS loss fixed ($SE+\mathrm{KGAT}+l_{\mathrm{CTS}}$) increases it by only 7. On CFD, the variant without multigrid even achieves slightly lower fill-in than the multigrid version.
>
> **5. On the choice of Fourier basis instead of B-splines in KAN.**
> In our setting, the model learns node scores for ordering under local triplet constraints, where small structural changes can induce non-smooth changes in the preferred elimination order. This makes the target signal inherently non-smooth and multi-scale. Fourier basis functions are well suited to the non-smooth, multi-scale structural patterns in our task and are also compatible with our multigrid backbone. In our setting, the Fourier parameterization was straightforward to integrate and showed better empirical training stability.
>
> **6. On multiple paths / intermediate neighbors between a non-adjacent pair.**
> This point is central to our formulation. A key contribution of the paper is to reduce the Fill-Path theorem from arbitrary-length paths between a non-adjacent pair to local three-node structures, and to prove a strict equivalence between global no-fill and local triplet blocking. If there exists any path satisfying the Fill-Path condition, then a fill edge will be created; equivalently, if all such candidate paths are blocked, then no fill edge is created. CTS is built exactly on this reformulation.
>
> **7. On the training-to-inference cost ratio.**
> For CTS, the additional cost is concentrated in offline training rather than inference.
> The training cost includes one pass of forward computation, triplet loss computation including triplet sampling and one pass of backward computation. The inference cost is only one pass of forward computation. Empirically, we report both costs below on the same set as Table 3. The training-to-inference cost ratio is approximately 3.
>
> | Matrix Name | Train Cost (MFlops) | Inference Cost (MFlops) | Train/Infer Ratio |
> |---|---:|---:|---:|
> | lhr10  | 256.11 | 85.37  | 3 |
> | lhr11  | 263.06 | 87.69  | 3 |
> | bayer10 | 317.99 | 106.00 | 3 |
> | lhr10c | 255.41 | 85.14  | 3 |
> | lhr14c | 341.67 | 113.89 | 3 |
> | lhr17  | 421.14 | 140.38 | 3 |
> | lhr17c | 419.70 | 139.90 | 3 |
> | lhr34c | 841.44 | 280.48 | 3 |
> | lhr11c | 262.07 | 87.36  | 3 |
>
> **8. On numerical stability.**
> CTS addresses the structural reordering stage and does not explicitly model numerical pivoting. Therefore, like the other graph-based reorderers in our comparison, it does not by itself guarantee numerical stability. Numerical pivoting and stability are handled later by the downstream factorization procedure.

---

> > ### Author Rebuttal · Reviewer_L5ce · 2026-04-04
> >
> > Thank you for the insightful rebuttal

---

> > > ### Author Response · Authors · 2026-04-07
> > >
> > > Thank you for your time and for carefully considering our rebuttal. We appreciate your thoughtful feedback.

---

### Official Review · Reviewer_YK6j · 2026-03-14

**Soundness:** 3
**Presentation:** 2
**Significance:** 2
**Originality:** 2
**Overall Recommendation:** 3
**Confidence:** 2

**Summary:**

This paper investigates sparse matrix reordering for LU fill-in reduction and proposes Causal Triplet Structure Learning (CTS). The main idea is to use the Fill-Path Theorem to reduce arbitrary fill-path reasoning to local candidate triplets, then train a multigrid graph-attention model with Fourier/KAN activations to predict ordering scores under a causal triplet loss that discourages structures expected to induce fill-in. On SuiteSparse benchmark subsets and additional CPS/CFD analyses, the paper reports the best mean fill-in ratios overall and favorable LU factorization times relative to classical reorderers and prior learning-based methods.

**Compliance With Llm Reviewing Policy:**

Affirmed.

**Key Questions For Authors:**

- The formal reduction is stated for symmetric sparsity. How are nonsymmetric matrices handled in practice when constructing the graph and applying CTS to the benchmark collection?

- Were the learning-based methods trained and evaluated over multiple random seeds? If so, can the authors report seed-level variability for the main Table 1 results?

- Will the authors release code, benchmark splits, and exact training configurations for CTS and the compared learning-based baselines?

- How were CTS hyperparameters selected relative to the baselines, and was any validation budget used consistently across methods?

- For the LU factorization time comparisons, how many runs were used per matrix and what steps were taken to control for runtime noise?

**Limitations:**

The impact statement does not seriously address scope, risks, unintended consequences.

**Strengths And Weaknesses:**

Strengths

- The paper tackles an important and practically relevant problem where even moderate gains can affect runtime and memory use in scientific computing.
- The methodological story is more formal than a typical heuristic reordering paper, and the connection to the Fill-Path Theorem gives the approach a clear conceptual backbone.
- The baseline suite is broad and relevant, covering natural ordering, classical graph-theoretic reorderers, and multiple learning-based methods.
- The empirical section is convincing to the main claims.

Weaknesses

- The paper risks overstating the novelty of the causal framing. What appears genuinely new is the local triplet reduction and its optimization objective.
- Statistical robustness is only moderate. The paper reports means and standard deviations across benchmark matrices, but not training-seed variability or a stronger runtime-measurement protocol for the reported speedups.
- Reproducibility is incomplete. Several architectural settings are given, but I do not see an anonymous code release or a compact reproducibility summary that would make the full pipeline easy to reimplement.
- The formal development appears to rely on symmetric sparsity, and the paper should explain more explicitly how that assumption maps onto the full benchmark setting in practice.
- The impact statement is too weak for ICML standards and does not seriously address scope, risks, unintended consequences.

Comments
- The main empirical conclusions are clearly stated. The paper argues that CTS achieves the best mean fill-in ratio across the evaluated problem groups, with especially strong results on CPS and CFD, and that these improvements usually translate into faster LU factorization.
- Reporting standard deviations across matrices is useful, but the variance across problem instances is large, runtime is explicitly hardware-sensitive, and I do not see seed-level variability for the learned methods.
- The generalization and scalability story is interesting but should be bounded carefully. Training on matrices of size 200 to 500 and testing on matrices larger than 10,000, including groups up to 15 million nonzeros, supports within-domain scale transfer more than broad out-of-domain generalization.
- The ablation and fill-pattern-overlap analyses are helpful, but they still do not fully isolate how much of the gain comes from the triplet objective versus the architecture, capacity, or tuning choices.

---

> ### Author Rebuttal · Authors · 2026-03-31
>
> We thank the reviewer for the thoughtful comments and constructive suggestions.
>
> **1. On the novelty of the “causal” framing.**
> We agree that the core methodological novelty of our work lies primarily in the local triplet reduction and its corresponding optimization objective. Here the term *causal structure* is not intended to refer to causal inference in the standard machine learning sense. Rather, it describes the cause-effect relationship between the elimination order within a triplet and the occurrence of fill-in: specifically, whether a fill edge is created is determined by the relative elimination order of the three nodes in the triplet. Compared with the direct optimization of fill-in effect as PFM, CTS attempts to reduce the causes of fill-ins. We will revise the paper more precise and avoid possible overstatement.
>
> **2. On the robustness to random seed.**
> The major randomness of CTS comes from the network weight initialization and the generation of training matrices. The performances reported in the paper are obtained with a heuristically fixed seed, without tuning. Its specific setting will be provided in the code release upon acceptance. To investigate the effect of random seed on the prediction performance, we test five seeds of different scales ranging from 10 to 10^6 and obtain the fill-in ratios as follows. The performance deviation is relatively small compared to the mean.
>
> | Matrix Name | Mean | Stdev |
> |---|---:|---:|
> | lhr10  | 112.48 | 0.93 |
> | lhr14c | 157.88 | 3.27 |
> | lhr11c | 105.96 | 1.06 |
>
> **3. On code and reproducibility.**
> Due to the rebuttal policy, we are unable to provide a code link at this stage. However, we plan to publicly release the full implementation in the final version, including the CTS codebase, benchmark splits, exact training configurations, and the implementations/configurations used for the learning-based baselines, to facilitate full reproducibility.
>
> **4. On nonsymmetric matrices.**
> Our formal development is presented for symmetric sparsity. In practice, for a nonsymmetric matrix \(A\), we first construct a symmetrized graph using \(A + A^\top\), obtain the ordering based on this graph, and then apply the resulting permutation back to the original matrix. This is also the standard practice followed by classical graph-based reorderers such as ND. We will clarify this implementation detail in the revision.
>
> **5. On hyperparameter selection and fairness to baselines.**
> For the baselines, we used the best hyperparameter settings reported in their original papers whenever available. For fairness, CTS and the learning-based baselines were trained and tuned to a comparable extent, and key architectural factors such as network depth and hidden dimensionality were kept aligned as much as possible across methods. We will add a clearer description of the tuning protocol in the revision.
>
> **6. On LU factorization time measurement.**
> For each ordering method, we performed one warm-up run before timing, and then reported the average over three subsequent runs. During evaluation, no other user programs were running, in order to reduce system interference as much as possible. We will make this runtime measurement protocol explicit in the paper.
>
> **7. On the impact statement.**
> Our work aims to reduce fill-in during sparse matrix factorization through learned reordering, and thus falls within the broader AI for HPC setting. By reducing unnecessary computation in sparse factorization, the method has the potential to improve computational efficiency and, consequently, reduce energy consumption in large-scale scientific computing workloads. We will strengthen the impact statement accordingly.

---

### Decision · Program_Chairs · 2026-04-30

**Decision:**

Accept (regular)

**Comment:**

This paper investigates sparse matrix reordering for LU fill-in reduction and proposes Causal Triplet Structure Learning (CTS). The main idea is to use the Fill-Path Theorem to reduce arbitrary fill-path reasoning to local candidate triplets, then train a multigrid graph-attention model with Fourier/KAN activations to predict ordering scores under a causal triplet loss that discourages structures expected to induce fill-in.

Empirical evaluation on the SuiteSparse Matrix Collection focuses on Chemical Process Simulation (CPS) and Computational Fluid Dynamics (CFD) matrices. The authors report up to a 6x reduction in fill-in ratios and a corresponding 12x acceleration in LU factorization time compared to established deep learning baselines like PFM and UDNO. The framework is designed to generalize from small training instances to matrices with over 1,000,000 rows by leveraging the local nature of the triplet constraints.

Reviewers agree that the paper tackles an important and practically relevant problem where even modest gains can be extremely important for downstream applicatinos. The methodological story is more formal than a typical heuristic reordering paper, and the connection to the Fill-Path Theorem gives the approach a clear conceptual backbone. Reviewers praise the paper for its experimental validation, which compares against a broad and relevant baseline suite, and provides convincing evidence to support its main claims.

Main criticism is on the novelty of the paper. Reviewers point out that the triplet reduction seems closely related to classical sparse elimination / perfect-elimination characterizations, so the true novelty is less clear and perhaps overstated. JMVP contests the “causal” framing, as while the paper identifies a structural condition for fill generation, but this is not causal inference in the usual machine learning sense. LZtc write that the methodology is more of a combination of existing techniques rather than a novel conceptual contribution.

In my reading, I find that the paper is well-written, comprehensive, and delivers what it promises. Despite the concerns with novelty, reviewers do not contest the empirical improvements and agree that the evidence supports the claims. The fundamental challenge is that the paper studies an already-mature problem (most low-hanging fruits were plucked by the 1980s) that is nevertheless extremely high-impact, where even marginal improvements could yield concrete benefits. It is an area where theoretical and practical improvements are difficult, and as such the engineering value and practical utility should not be overlooked. This paper has value as a stepping stone in the emerging field of AI+SciComp, and with public release of the code, could have immediate and direct impact on downstream applications.

For the final version, the authors should "publicly release the full implementation in the final version, including the CTS codebase, benchmark splits, exact training configurations, and the implementations/configurations used for the learning-based baselines, to facilitate full reproducibility," as they promised in their rebuttal to YK6j.